# Autonomous Manipulator of a Mobile Robot Based on a Vision System

Anna Annusewicz-Mistal [1], Dawid Sebastian Pietrala [1], Pawel Andrzej Laski [1,*], Jaroslaw Zwierzchowski [2], Krzysztof Borkowski [1], Gabriel Bracha [1], Kamil Borycki [1], Szczepan Kostecki [1] and Daniel Wlodarczyk [1]

[1] Faculty of Mechatronics and Mechanical Engineering, Kielce University of Technology, Aleja Tysiaclecia Panstwa Polskiego 7, 25-314 Kielce, Poland

[2] Department of Microelectronics and Computer Science—DMCS, Lodz University of Technology, 90-924 Lodz, Poland

\* Correspondence: pawell@tu.kielce.pl; Tel.: +48-41-3424504

**Featured Application: The autonomous manipulator could be widely used for manipulation tasks such as handling panels or grasping and moving objects without humans who control the robot.**

**Abstract:** This article presents a system for the autonomous operation of a manipulator of a mobile robot. The aim of the research was to develop a system for a robot to operate a panel with switches. The manipulator should position itself autonomously and perform a given action. The operation of the system is based solely on one digital camera. The vision system uses markers to identify the position of the end-effector. The test results show that the system operates well in good artificial and natural lighting conditions. The system can be used effectively for activities that do not require high accuracy, e.g., pressing buttons and switches or grasping objects. However, for high-precision tasks, such as inserting a plug into a socket, or tasks that require high positioning accuracy, the manipulator may not be suitable.

**Keywords:** manipulator; vision system; ARTag; mobile robot

## 1. Introduction

Recent technological advancements allow robotic systems to be more and more autonomous. Much research in the field of autonomous robots is now focusing on autonomous navigation in specific environments or autonomous positioning to perform specific tasks, for example, to grip an object. Improvements in this area are mainly due to the continued innovation in computer vision. The most common solution is to use fiducial markers to estimate the camera pose within an environment in real time.

Augmented reality (AR) fiducial markers are widely used in robotics as they are suitable for navigation and positioning purposes. The most common are ARTag [1], ARToolkit [2], and ArUco [3]. The major application of marker-based vision systems is to navigate mobile robots. Knowing the position of the markers on the map, a robot is able to locate itself in space. Another approach assumes that markers can indicate the destination the robot is required to reach.

The research presented in [4] involved using ARTag markers to navigate a mobile robot. They helped increase the accuracy of GPS data. Another study concerned with the navigation of mobile robots [5] aimed to develop an algorithm for fusing the wheel odometry and IMU data and a vision system capable of detecting ARTags, estimating the distances to them, and correcting the robot's position determined from the IMU data. The use of marker-based vision systems for robot navigation is also discussed in [6–8].

Artificial markers can be employed for the navigation and positioning of unmanned aerial vehicles (UAVs). As indicated in [9], the localization of multirotor aerial vehicles

used in indoor applications can be based on visual inertial odometry and a map of markers. In the study of the localization of micro air vehicles (MAVs) [10], the researchers propose fusing the data from sensors with those provided by the vision system using ArUco markers. AR markers were also used in research on the autonomous landing of unmanned aerial vehicles at the desired location, including a mobile platform [11–13].

Marker-based vision systems are also used to assist in manipulation and positioning. As suggested in [14], a marker tracking system can be integrated into microelectromechanical systems (MEMS) and similar devices to provide real-time data on the position of microscale objects. The research on multirobot industrial systems, e.g., [15], shows that ArUco markers can improve the interaction between a fixed (stationary) robot and a mobile robot, ensuring high-accuracy positioning of the manipulator and high-accuracy selection of details from the mobile platform. As indicated in [16], a manipulator equipped with a marker and camera-based system can accurately position the end-effector to grasp an object and move it to the desired position. The vision system proposed in [17] uses markers to control the position and orientation of excavator arms (booms). The paper [18] presents a system for estimating manipulator positions using ArUco tags.

The aim of the study described in this article was to develop an autonomous panel operation system for a mobile robot equipped with a manipulator. In this article, the term robot is used to refer to a platform to which the manipulator is attached. It was assumed that the task performed by the robot would be a selected switch. The position of the manipulator is determined by detecting the positions of the markers located on the panel surface. By determining the locations of the switches on the panel in relation to the markers, the manipulator is able to run appropriate movement sequences to turn them on or off.

The two major subsystems responsible for the manipulator's performance are the vision and control systems. The vision system first detects the markers and then calculates the position and orientation of the end-effector with respect to the panel. The control system uses the data to move the end-effector to a desired position in the coordinate system so that the task can be performed with the highest accuracy possible. Another system incorporated into the unmanned ground vehicle (UGV) is the autonomous manipulator control system. A vehicle with a total weight of 50 kg was designed, constructed, and programmed by the IMPULS student research group affiliated with the Department of Automation and Robotics of the Kielce University of Technology in Poland. The four-wheeled vehicle is equipped with a specifically developed six-degree-of-freedom manipulator and a wireless communication and control system. The vehicle navigation and manipulator operation are controlled by STM microprocessors programmed, for instance, to solve the inverse kinematics of the manipulator.

The IMPULS robot project involved (a) developing vision system algorithms to determine the position of elements on the operator panel in relation to the end-effector, (b) installing the vision system hardware, i.e., a camera and a minicomputer, (c) analyzing the manipulator positioning accuracy, (d) assessing the marker recognition accuracy, and (e) determining the accuracy of the autonomous positioning of the end-effector relative to the markers.

Section 2 of this article describes the design and control of the manipulator; it also includes the algorithm for the vision system. Section 3 provides the research results, which are discussed in Section 4. The last section outlines the key conclusions drawn, as well as proposals for the further development of the robotic system. The improved manipulator is expected to be able to work alongside humans as a collaborative robot, performing simple tasks such as the preparation of beverages in the catering services sector or performing pick-and-place operations along with object orientation detection in the manufacturing sector.

## 2. System Description

The autonomous system was used in a mobile robot equipped with a manipulator, shown in Figure 1.

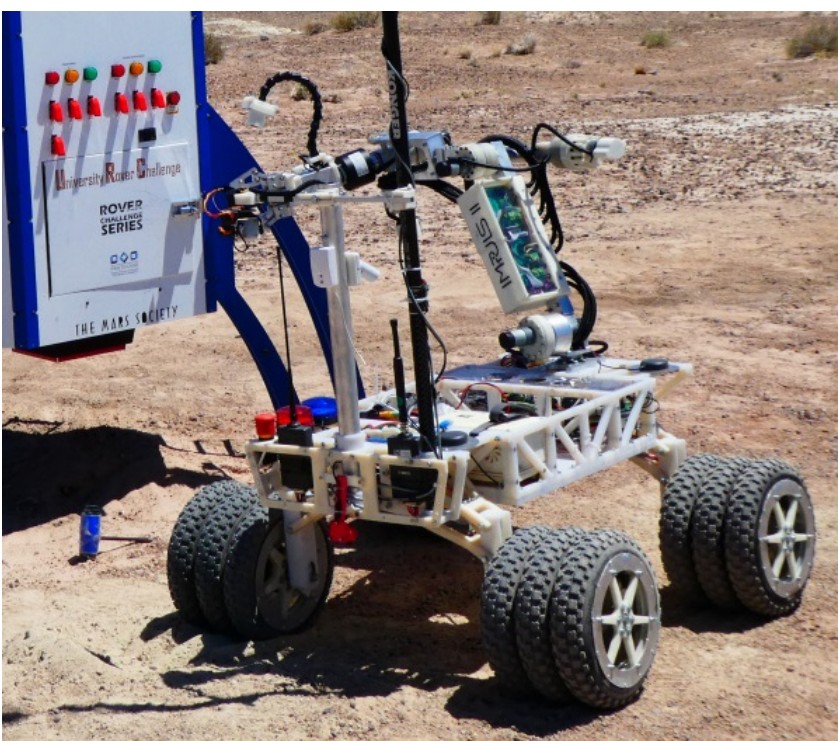

**Figure 1.** Mobile robot during analog simulation mission.

### 2.1. Manipulator

The manipulator has six degrees of freedom. It is composed of six 5th-class rotary kinematic pairs. As a drive, it uses DC motors with planetary gear with a ratio from 1:30 to 1:70, depending on the axis. For each axis, an additional gear was used in the form of a harmonic drive with a gear ratio of 1:160. This allowed for high positioning accuracy and torque. Incremental encoders with a resolution of 64 pulses per revolution were used to measure the axis rotation. Taking into account the full gear ratio, the obtained values for the respective axis are from 307,200 to 716,800 pulses per revolution.

A kinematic analysis was performed for the manipulator using the Denavit–Hartenberg notation [19]. The base coordinate system is located in the base of the robot. The coordinate system of each axis is defined relative to the coordinate system of the previous axis by a matrix of $4 \times 4$ dimension. The matrix $M$ shows the general form of the transformation (1). The matrix $R$ is a $3 \times 3$ dimension rotation matrix, the vector $t = [t_x, t_y, t_z]^T$ is a translation column vector, and $0^T = [0, 0, 0]$.

$$M = \begin{bmatrix} R & t \\ 0^T & 1 \end{bmatrix}, \tag{1}$$

The transformation matrix is created by combining homogeneous transformations: Translations and rotations around the $x$, $y$, and $z$ axes. Multiplying the transformation matrices provides the position and orientation of the last system with respect to the base system. The derivation of the manipulator kinematics equations is presented in Appendix A.

The control system consists of a motherboard equipped with an STM32F407VGT6 microcontroller and four dual servo drivers consisting of STM32F407 microcontrollers. Each controller is connected to a dual VNH5019 integrated circuit that controls the power for two DC motors and two incremental encoders. The servo drivers control the angular position with the use of PID regulators, and the motherboard carries out the manipulator kinematics algorithms. Three servo controllers operate the six main drives of the manipulator, while the fourth servo controller operates the gripper drives.

### 2.2. Vision System

The vision system uses the Logitech HD Pro Webcam C920 digital camera with FullHD resolution (1920 × 1080 pixels) and a field of view of 78 degrees. The algorithm was implemented on the NVIDIA Jetson TX2 microcomputer. The camera is mounted on the gripper. As a result, the position of the gripper is determined by determining the position of the camera in relation to the panel.

The task of the vision system is to determine the position of the end effector in relation to the panel. On this basis, the algorithm determines the value and direction of movement that the manipulator should perform in order to obtain the given position.

ARTag markers [1] were used to estimate the position of the camera in relation to the panel. The test panel with markers is shown in Figure 2.

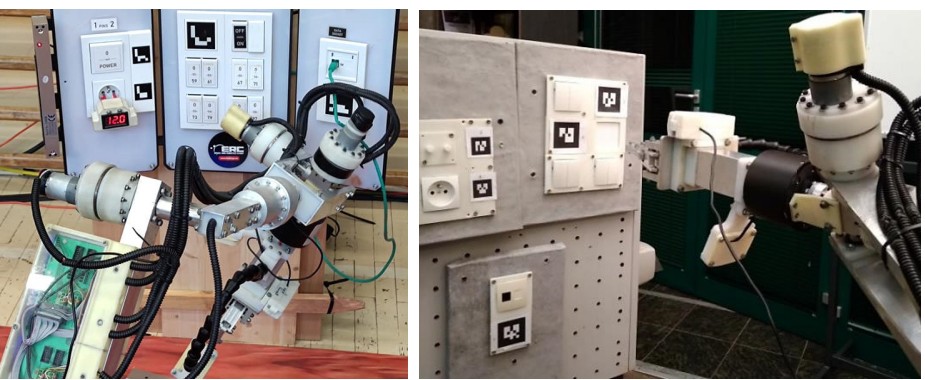

**Figure 2.** Test panel with ARTag markers.

### 2.3. Marker Detection Algorithm

The algorithm and methods presented in [3,14,20] were used to estimate the position. This algorithm allows us to identify markers of various types that have a frame and an internal code of a fixed size, e.g., Aruco or ARTag.

The image analysis for marker detection starts with converting the image to grayscale. Then, adaptive thresholding is performed in order to obtain the edges. Next, the algorithm searches for contours [21] and approximates them by means of a polygon [22]. The algorithm searches only for contours that meet the assumptions. The markers are square, so their image will be a convex quadrilateral. The remaining contours are discarded. The perspective view is removed from the image of the contours that remain after filtration so that the code can be read. A marker is considered correct if it has a black frame and a code compatible with the codes written in the program. To check this, thresholding [23] is carried out.

For each detected marker, the position and orientation of the camera relative to the system associated with the marker are computed. The method of solving the Perspective-n-Point problem using the Levenberg–Marquardt optimization [24] was used to estimate the position. Upon obtaining a position in the global coordinate system in 3D space for at least four points and the projection coordinates of these points, it is possible to determine the orientation and position of the camera in relation to the global system. For exactly four points, the solution is unambiguous, and in the case of a greater number of points, the result is optimized with the chosen method.

The algorithm was implemented in C++ using the OpenCV and QT libraries on the NVIDIA microcomputer.

Figure 3 shows the image from the camera while positioning the manipulator.

In order to identify the internal parameters of the camera, a calibration process was performed for a fixed sharpness value. The camera was calibrated using the Zhang method [25] in Matlab.

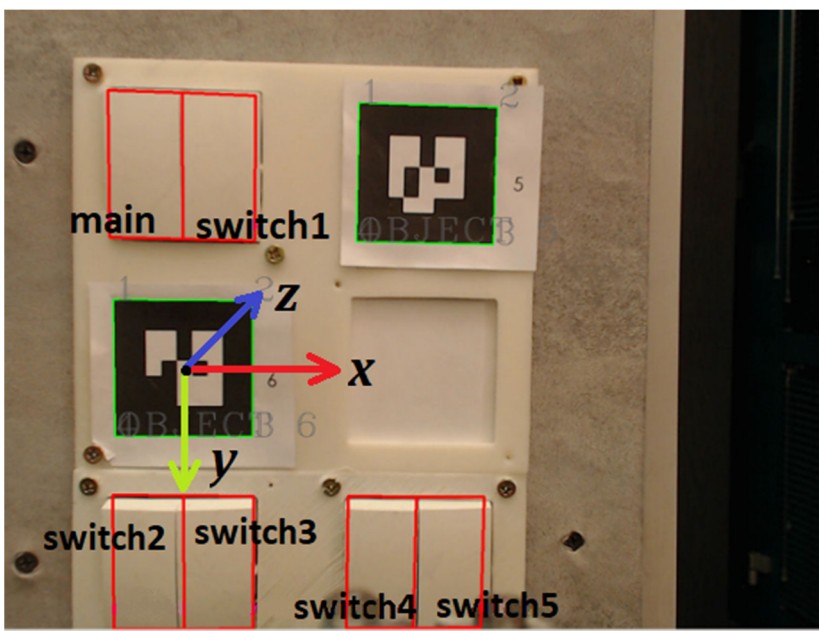

**Figure 3.** View from the camera mounted on the gripper with recognized markers and marked switches.

### 2.4. Positioning Based on a Vision System

The system was developed with the assumption that the position of the appropriate markers on the panel and the location and type of objects are known. In the panel shown in Figure 3, there are two different markers, and the center of the left marker has the global coordinate system. In this case, the maximum number of points for estimation position is eight, so it increases the accuracy.

The operation of the system begins with searching for markers and determining the position of the gripper relative to the coordinate system on the panel, as shown in Figure 4. When the cycle starts, the operator does not interfere with the robot's work. The starting position of the manipulator is arbitrary, provided that at least one marker in the frame is visible. After determining the position, we calculated how much the robot must move in each axis of the tool coordinate system and how much it should rotate in relation to these axes in order to position itself at a base point in front of the panel. When the intended position is reached, the markers are detected again and the position relative to which the next movements are determined is calculated. Before the robot completes the task, it successively arrives at several predetermined points where the position is calculated. It increases the accuracy because the farther the camera is from the markers, the less accurately the position is computed. At intermediate points, previous errors are eliminated. The last measurement is performed just in front of the object so that the marker is visible. Marker detection only takes place at fixed points while the robot does not move. Thanks to this, the image is of good quality and the movements between the points do not have to have speed limits. The algorithm of marker detection is shown in Algorithm 1. There are several necessary steps starting from image conversion (color version to greyscale version), passing throw marker contour detection (only rectangle shapes are allowed), and ending with checking coded in marker number and numbers database stored in manipulator memory. In the case of determining the manipulator distance from the marker, we used the algorithm and the fact that the marker frame has known dimensions and shape. 2D image points and their 3D correspondences are taken only from the outer edges of the marker frame. We also check the thickness of the marker frame and its code. The positions of the markers located on the panel are known relative to the global coordinate system, so in the PnP algorithm, all detected marker points could be used.

| **Algorithm 1.** Marker detection algorithm. |
| --- |
| **Input**: images with possible markers<br>**Output**: markers with code (number) and manipulator positions |
| 1.    Convert image to greyscale<br>2.    Detect edges<br>3.    Detect contours, **if** contour too small **than** remove<br>4.    Approximate contours with polygons<br>5.    Filter contours, **if** not 4 sides of equal lengths **than** remove<br>6.    Remove perspective projections (only windows with detected marker candidates)<br>7.    Save all image marker candidates<br>8.    **If** marker detected **than** read marker code **else** go to line 10<br>9.    **If** code in robot database **than** save and calculate manipulator position **else** go to line 10<br>10.  **If** not all candidate **than** go to line 8 |

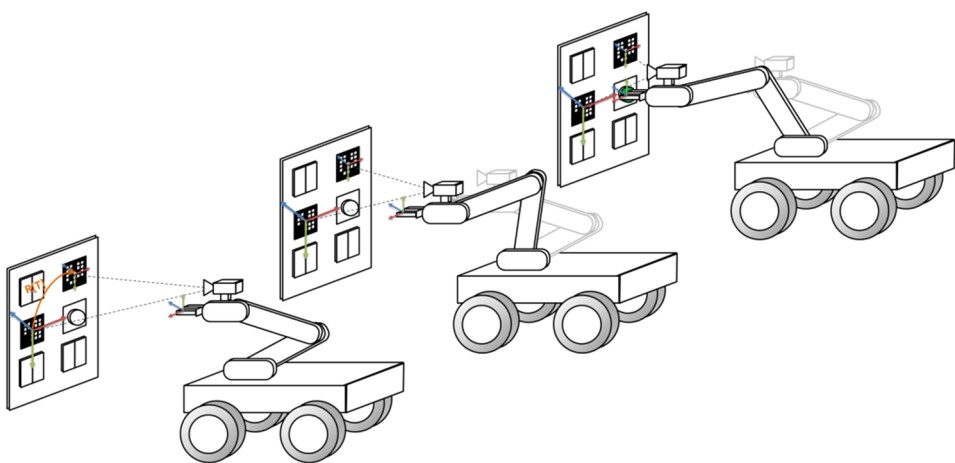

**Figure 4.** The manipulator adjusts its gripper coordinate system to the panel's global coordinate system located in one of the recognized markers.

If no markers are detected at an intermediate point, the robot will move relative to the previously calculated position.

## 3. Results

To check and evaluate the operation of the system, three tests were carried out. The first examined the execution accuracy of the manipulator, the second explored the accuracy of the vision system in estimating the position of the camera relative to the marker, and the third assessed the manipulator positioning based on the operation of the vision system.

### 3.1. Manipulator Accuracy Testing

The test consisted of setting the distance by which the manipulator should move for the selected axis and measuring how much it actually shifted. Measurements were made for selected axes. The operation has been checked for various values, including 20, 50, and 100 mm. In total, more than 60 measurements were made. Table 1 shows the measurements for the case of displacements for the *z*-axis. The basic statistical parameters were calculated on the basis of the measurements. The actual manipulator movement was obtained simply, as we had set some linear movement to the manipulator control system and then measured where the end effector moved. The position value was taken from a precision plunger dial indicator from the RS Pro company.

**Table 1.** Measurements of the manipulator's z-axis movements for a setpoint of 50 mm.

| Actual Manipulator Movement mm | Absolute Error mm | Relative Error |
|---|---|---|
| 49.9 | 0.1 | 0.20% |
| 49.8 | 0.2 | 0.40% |
| 50 | 0 | 0.00% |
| 49.7 | 0.3 | 0.60% |
| 49.7 | 0.3 | 0.60% |
| 50 | 0 | 0.00% |
| 51.9 | 1.9 | 3.80% |
| 47 | 3 | 6.00% |
| 49.3 | 0.7 | 1.40% |
| 49.4 | 0.6 | 1.20% |

*3.2. Vision System Accuracy Testing*

This section shows the measurement of the camera position relative to the marker. The measurements were performed as a function of the distance from the marker and the tilt angle in relation to the selected axis because such changes have a significant impact on the accuracy of the vision system. In this test, the camera was mounted on an IRB 1600 ABB industrial robot because the industrial robot is highly accurate (according to the robot specification, the robot pose accuracy is 0.04 mm) [26]. Its position in relation to the marker system was taken as the reference position. Tables 2 and 3 show the obtained results, and Figure 5 shows a diagram of the dependence of the error on the distance to the marker. The tests were carried out under good lighting conditions. Artificial laboratory lighting and a 250 W lamp directly illuminating the panel were used.

**Table 2.** Distance measurements by a vision system.

| Reference Distance mm | Vision System mm | Absolute Error mm | Relative Error |
|---|---|---|---|
| 200.1 | 200.5 | 0.4 | 0.20% |
| 218.3 | 219.2 | 0.9 | 0.41% |
| 236.8 | 237.6 | 0.8 | 0.34% |
| 255.6 | 256.3 | 0.7 | 0.27% |
| 274.5 | 275.3 | 0.8 | 0.29% |
| 293.6 | 294.5 | 0.9 | 0.31% |
| 312.8 | 313.7 | 0.9 | 0.29% |
| 332 | 332.3 | 0.3 | 0.09% |
| 351.4 | 351.7 | 0.3 | 0.09% |
| 370.8 | 371.8 | 1 | 0.27% |
| 390.3 | 390.6 | 0.3 | 0.08% |
| 409.9 | 410 | 0.1 | 0.02% |
| 429.4 | 429.6 | 0.2 | 0.05% |
| 192.4 | 192.5 | 0.1 | 0.05% |
| 208.4 | 209 | 0.6 | 0.29% |
| 225.5 | 225.9 | 0.4 | 0.18% |
| 243.5 | 243.6 | 0.1 | 0.04% |
| 262.1 | 262.1 | 0 | 0.00% |
| 281.3 | 281.1 | 0.2 | 0.07% |
| 300.9 | 300.6 | 0.3 | 0.10% |
| 320.9 | 320.3 | 0.6 | 0.19% |
| 341.2 | 338.6 | 2.6 | 0.76% |
| 361.7 | 359.1 | 2.6 | 0.72% |
| 382.5 | 379.3 | 3.2 | 0.84% |
| 403.4 | 397.5 | 5.9 | 1.46% |
| 424.4 | 421.3 | 3.1 | 0.73% |

**Table 3.** Measurement of rotation around the *y*-axis by a vision system.

| Reference Angle Deg | Vision System Deg | Absolute Error Deg |
|---|---|---|
| 0 | 0.4 | 0.4 |
| 0 | 3.19 | 3.19 |
| 0 | 3.33 | 3.33 |
| 0 | 3.4 | 3.4 |
| 0 | 3.36 | 3.36 |
| 0 | 3.53 | 3.53 |
| 0 | 3.47 | 3.47 |
| 0 | 3.45 | 3.45 |
| 0 | 3.71 | 3.71 |
| 0 | 0.31 | 0.31 |
| 0 | 3.54 | 3.54 |
| 0 | 3.64 | 3.64 |
| 0 | 0.74 | 0.74 |
| 4 | 4.36 | 0.37 |
| 6 | 8.21 | 2.21 |
| 8 | 9.98 | 1.98 |
| 10 | 11.65 | 1.65 |
| 12 | 13.44 | 1.44 |
| 14 | 15.32 | 1.32 |
| 16 | 16.98 | 0.98 |
| 18 | 18.86 | 0.86 |
| 20 | 20.06 | 0.06 |
| 22 | 22.56 | 0.56 |
| 24 | 21.73 | 2.27 |
| 26 | 23.17 | 2.83 |

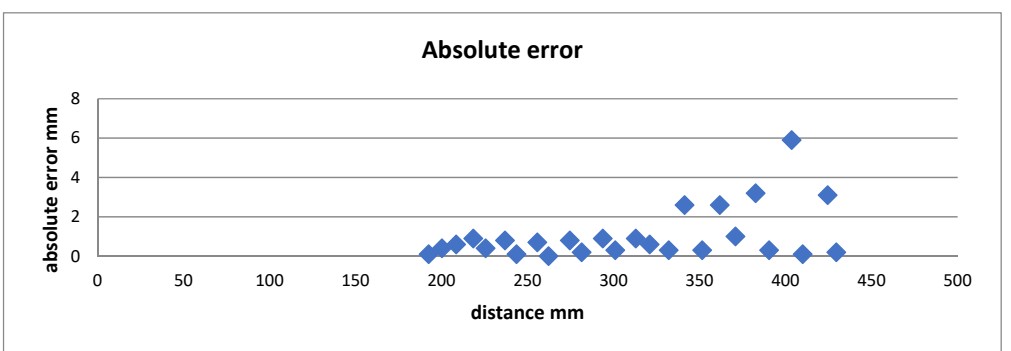

**Figure 5.** The dependence of the measurement error on the distance from the marker.

### 3.3. Testing the Positioning System on the Basis of the Vision System

The last tests concern combining the operation of both systems. A positioning test was carried out for the assumed base position. The manipulator should be in the same position every time. The vision system has a greater impact on errors, and its accuracy varies with the distance and angle of inclination to the panel. Table 4 shows the positions relative to the manipulator's base coordinate system reached by the end effector and the actual distance from the panel. During these tests, the flexible suspension of the robot was stiffened so that the movement of the entire robot did not affect the position of the manipulator effector. The set base position was 170 mm from the panel. This test allowed us to evaluate the positioning accuracy based on the vision system. The manipulator reached positions with an error of several millimeters, and this is due to the different initial positions from which the vision system made the measurement. As shown by previous tests, the greater the measurement distance and the greater the angle of the camera relative to the marker, the more inaccurate the measurements. The actual distances from the panel were measured using the laser sensor di-soric LAT 51 M 500 IG3-B5.

**Table 4.** Measurement of the distance from panel to the manipulator in the base position obtained by the manipulator for a given distance of 170 mm.

| Position Reached in $x$, $y$, $z$ mm | | | Actual Distance from the Panel mm | Absolute Error of Distance mm |
|---|---|---|---|---|
| 577.1 | −319.0 | 587.4 | 162.1 | 7.9 |
| 590.0 | −312.0 | 575.7 | 153.6 | 16.4 |
| 575.1 | −324.4 | 587.5 | 161.2 | 8.8 |
| 566.6 | −334.8 | 593.0 | 166.5 | 3.5 |
| 552.9 | −334.0 | 599.1 | 175.2 | 5.2 |
| 560.3 | −337.2 | 593.3 | 170.1 | 0.1 |
| 562.4 | −336.3 | 593.9 | 171.3 | 1.3 |
| 558.1 | −334.2 | 596.2 | 175.1 | 5.1 |
| 559.3 | −341.5 | 595.0 | 170.2 | 0.2 |
| 570.0 | −324.5 | 590.4 | 165.8 | 4.2 |
| 571.7 | −325.2 | 589.8 | 164.7 | 5.3 |
| 566.7 | −328.0 | 593.1 | 168.3 | 1.7 |
| 576.6 | −318.8 | 586.9 | 161.5 | 8.5 |
| 559.5 | −335.7 | 596.9 | 171.3 | 1.3 |
| 560.4 | −335.8 | 598.4 | 171.5 | 1.5 |
| 567.3 | −330.6 | 594.0 | 168.3 | 1.7 |

Another test was performed to determine whether the task to be performed was successful, i.e., switching the switch. In this case, the position was measured after pressing the switch. Moreover, in this case, the reference position was read relative to the manipulator's base coordinate system.

During the tests, the markers were detected at each point in the sequence. The operation of the system was also checked when it was impossible to calculate the correction. The results are given in Tables 5 and 6.

**Table 5.** Measurement of the position when switching the switch. The reference position is 747.1, −262.1, 544.1 mm.

| Position Reached in $x$, $y$, $z$ mm | | | Absolute Error for the $x$, $y$, $z$ Axes mm | | |
|---|---|---|---|---|---|
| 754.6 | −270.1 | 543 | 7.5 | 8 | 1.1 |
| 758.1 | −269.1 | 541.2 | 11 | 7 | 2.9 |
| 759.2 | −269.2 | 539 | 12.1 | 7.1 | 5.1 |
| 756.2 | −267.8 | 540.7 | 9.1 | 5.7 | 3.4 |
| 755.9 | −267.7 | 540.5 | 8.8 | 5.6 | 3.6 |
| 756.4 | −266.9 | 540 | 9.3 | 4.8 | 4.1 |
| 755.2 | −266.4 | 540 | 8.1 | 4.3 | 4.1 |
| 754.8 | −266.7 | 539.2 | 7.7 | 4.6 | 4.9 |
| 754.2 | −260 | 546 | 7.1 | 2.1 | 1.9 |
| 755.5 | −262.8 | 542.4 | 8.4 | 0.7 | 1.7 |

**Table 6.** Measurement of the position when switching the switch in the case of covering the markers during the work. The reference position is 747.1, −262.1, 544.1 mm.

| Position Reached in $x$, $y$, $z$ mm | | | Absolute Error for the $x$, $y$, $z$ Axes mm | | |
|---|---|---|---|---|---|
| 705.5 | −266.5 | 568.4 | 41.6 | 4.4 | 24.3 |
| 760.3 | −261.3 | 487.9 | 13.2 | 0.8 | 56.2 |

## 4. Discussion

Based on the research on the positioning of the manipulator, it was determined that in more than 90% of cases, the error is less than 1 mm. The standard deviation was 0.617 mm. Half of the errors obtained were below the value of 0.4 mm, and 75% were below the value of 0.6 mm. This means that the positioning of the manipulator works with good accuracy, sufficient for robot applications. The maximum error was 3 mm, and the minimum error was 0 mm.

The vision system tests have shown that the distance measurement is relatively accurate, and the measurement error is, in most cases, less than 1 mm. The relative error is small, in most cases less than 0.5%, and it becomes noticeably bigger for longer distances above 340 mm and is in the range of 0.7–1.5%. Measurements of rotation around the axis of the coordinate system are burdened with greater error, and they can even exceed 3°, which has a significant influence on the operation of the entire manipulator control system. The tests were performed at different distances from the marker and at different angles. For long distances above 340 mm, the error in measuring the distance could be much greater and could even be a few millimeters. The vision system settings allow the detection of a marker from a distance in the range of approximately 200–420 mm for a marker with a side of 50 mm. The error in measuring the distance grows as the angle of inclination relative to the marker increases. In Table 2, the first half of the measurements were made from a position opposite the panel; in this group, the maximum error was 0.97 mm and the minimum error was 0.13 mm. The second group consisted of measurements for an increasing angle. The unfavorable position resulted in a large error of 5.9 mm. For the performed measurements, the standard deviation was 1.69 mm. Half of the measurements had an error below 0.61 mm and 75% of the measurements were below 0.94 mm.

In other work, authors obtained similar results of position measurement using the vision system. However, in each study, the research was carried out on a different scale. For example, in the case of navigation, longer distances (several meters) were measured against larger markers (15–20 cm). An example can be seen in a previous study [20], where the authors obtained a relative error usually below 1%, and for long distances, below 2%. In another study [6], tests were carried out for a distance of up to 150 cm and a marker with a size of 100 mm. In a range of up to 1 m, the authors obtained an error close to zero for small distances, which increased to several millimeters with the increasing distance. Another study [14] examined the problem on the micro scale, and the obtained relative error did not exceed 1%. The most similar study was performed in the paper [18], in which the estimation of the manipulator position was studied. The authors obtained an average absolute error for position measurements using a vision system of 1.8 mm, 0.2 mm, and 3.5 mm for the $x$, $y$, and $z$ axes, and an average relative error of 1.6%, 0.2%, and 0.2%, respectively. Measurements were performed for a distance of 100–120 cm and a board with ArUco markers (A4 sheet of paper in size).

Based on the research on the operation of the combined system, it was observed that the positioning error was greater than it would appear from separate studies of the systems, which had much lower errors. This error is, on average, a few millimeters. It was influenced by the error in the rotation measurement, which was up to several degrees. The system was tested by starting the measurement at various distances ranging from 300 to 400 mm and at different angles to the panel, from a straight view to approximately 45° sweating. For larger camera angles and longer distances, the positioning error was greater. However, the operation of the system ended in success each time, i.e., a correctly switched button, because after reaching the base position in front of the panel, the position was counted from scratch, and corrections were introduced. The results of the successful tests are shown in Table 3. Table 4 shows the results of the failed tests. During the second test, the markers were covered after reaching the base position and the counting of corrections was impossible. This resulted in a big mistake, and the end of the task was unsuccessful.

## 5. Conclusions

The proposed manipulator positioning system based on the vision system underwent rigorous testing. It is noteworthy that the system was successfully used to perform all the tasks involving autonomous panel operation required for the European Rover Challenge 2021 and 2022 competition, simulating some aspects of missions to Mars.

Equipped with this autonomous positioning system, the manipulator is able to perform such tasks as pressing and releasing switches and buttons or turning knobs on the panel and picking up and placing objects. Because of the insufficient accuracy, the system is not suitable for high-precision operations, e.g., inserting a plug into a port or gripping very small objects.

Incorporating this autonomous system into an unmanned ground vehicle requires that corrections to the position of the manipulator should be made in real time. The shock-absorbing suspension and other elements of the vehicle structure are responsible for changes in the tilt and thus changes in the position of the center of gravity of the moving robotic arm. Failure to introduce corrections in real time may result in failure to perform the required task. The tests that involved locking the suspension (tilt) showed that the manipulator performed the task correctly. There were also tests performed with non-stiffened suspension, which gave correct results.

The resulting vision system error is very small if the manipulator is closer to the panel. In the case of a greater distance (above 340 mm), a larger error is not a problem, because after positioning the manipulator in front of the panel, corrections to the position are calculated in two places, in the base position and just in front of the switch. From the close proximity to the marker, the system calculates the position with high accuracy, which allows for accurate arrival and execution of the task, despite significant initial errors. In addition, the calculation of the correction is necessary due to the movable suspension of the entire robot. The results presented in the paper were obtained under laboratory conditions, with powerful artificial lighting (250 W lamp). However, the system also worked outdoors and indoors under daylight conditions and performed its function correctly.

Further research on the system will focus on increasing the autonomy, which will involve recognizing and localizing markers on the panel by using artificial neural network models to handle unknown panels.

**Author Contributions:** Conceptualization, A.A.-M., P.A.L., J.Z. and D.S.P.; methodology and software A.A.-M. and J.Z.; validation, K.B. (Krzysztof Borkowski), D.W., S.K., G.B. and K.B. (Kamil Borycki); formal analysis, investigation, and resources: A.A.-M., K.B. (Krzysztof Borkowski), D.S.P., K.B. (Kamil Borycki), G.B. and S.K.; writing—original draft preparation, P.A.L.; writing—review and editing, A.A.-M., P.A.L. and D.S.P.; visualization, A.A.-M., K.B. (Krzysztof Borkowski), K.B. (Kamil Borycki), S.K. and D.W. All authors have read and agreed to the published version of the manuscript.

**Funding:** Project co-financed Ministry of Education and Science No. SKN/SP/534769/2022.

**Institutional Review Board Statement:** Not applicable.

**Informed Consent Statement:** Not applicable.

**Data Availability Statement:** Not applicable.

**Conflicts of Interest:** The authors declare no conflict of interest.

## Appendix A

This appendix provides details on the solution of the manipulator kinematic analysis. The transformation matrix takes the general form:

$$M = \begin{bmatrix} R & t \\ \mathbf{0}^T & 1 \end{bmatrix} \tag{A1}$$

It can be decomposed into homogenous transformations of translation and rotation around the respective axes:

$$Trans_{x,y,z} = \begin{bmatrix} 1 & 0 & 0 & x \\ 0 & 1 & 0 & y \\ 0 & 0 & 1 & z \\ 0 & 0 & 0 & 1 \end{bmatrix}; \; Rot_{x,\alpha} = \begin{bmatrix} 1 & 0 & 0 & 0 \\ 0 & \cos\alpha & -\sin\alpha & 0 \\ 0 & \sin\alpha & \cos\alpha & 0 \\ 0 & 0 & 0 & 1 \end{bmatrix} \quad \text{(A2)}$$

$$Rot_{y,\beta} = \begin{bmatrix} \cos\beta & 0 & \sin\beta & 0 \\ 0 & 1 & 0 & 0 \\ -\sin\beta & 0 & \cos\beta & 0 \\ 0 & 0 & 0 & 1 \end{bmatrix}; \; Rot_{z,\gamma} = \begin{bmatrix} \cos\gamma & -\sin\gamma & 0 & 0 \\ \sin\gamma & \cos\gamma & 0 & 0 \\ 0 & 0 & 1 & 0 \\ 0 & 0 & 0 & 1 \end{bmatrix} \quad \text{(A3)}$$

The manipulator diagram is shown in Figure A1 with Denavit–Hartenberg parameters marked.

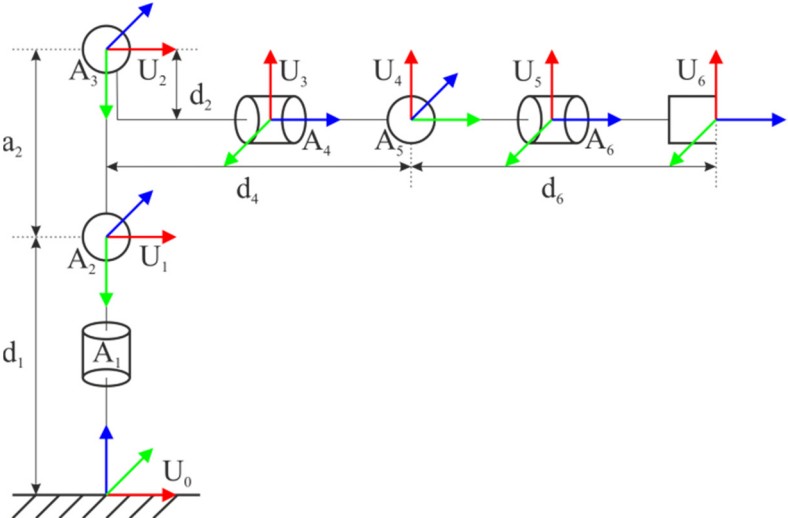

**Figure A1.** Kinematic diagram of the manipulator [27].

Table A1 shows the values of the D-H parameters; the variables $\theta_i$, where $i = 1, 2, \ldots 6$, are the configuration coordinates.

**Table A1.** Denavit–Hartenberg parameters.

| $i$ | $\theta_i$ [rad] | $d_i$ [mm] | $a_i$ [mm] | $\alpha_i$ [rad] |
|---|---|---|---|---|
| 1 | $\theta_1$ | 73 | 0 | $-\pi/2$ |
| 2 | $\theta_2$ | −98 | 400 | 0 |
| 3 | $\theta_3 - \pi/2$ | 0 | 0 | $-\pi/2$ |
| 4 | $\theta_4$ | 500.9 | 0 | $\pi/2$ |
| 5 | $\theta_5$ | 0 | 0 | $-\pi/2$ |
| 6 | $\theta_6$ | 190.9 | 0 | 0 |

The transformation matrices between successive coordinate systems of the manipulator parts are presented below.

$$M_1^0 = Rot_z\theta_1 \cdot Trans_z d_1 \cdot Trans_x a_1 \cdot Rot_x - \pi/2 = \begin{bmatrix} \cos\theta_1 & 0 & -\sin\theta_1 & a_1\cos\theta_1 \\ \sin\theta_1 & 0 & \cos\theta_1 & a_1\sin\theta_1 \\ 0 & -1 & 0 & d_1 \\ 0 & 0 & 0 & 1 \end{bmatrix} \quad \text{(A4)}$$

$$M_3^2 = Rot_z\theta_3 \cdot Trans_z d_3 \cdot Trans_x a_3 \cdot Rot_x - \pi/2 = \begin{bmatrix} \sin\theta_3 & 0 & \cos\theta_3 & a_3\sin\theta_3 \\ -\cos\theta_3 & 0 & \sin\theta_3 & -a_3\cos\theta_3 \\ 0 & -1 & 0 & d_3 \\ 0 & 0 & 0 & 1 \end{bmatrix} \tag{A5}$$

$$M_4^3 = Rot_z\theta_4 \cdot Trans_z d_4 \cdot Trans_x a_4 \cdot Rot_x \pi/2 = \begin{bmatrix} \cos\theta_4 & 0 & \sin\theta_4 & a_4\cos\theta_4 \\ \sin\theta_4 & 0 & -\cos\theta_4 & a_4\sin\theta_4 \\ 0 & 1 & 0 & d_4 \\ 0 & 0 & 0 & 1 \end{bmatrix} \tag{A6}$$

$$M_5^4 = Rot_z\theta_5 \cdot Trans_z d_5 \cdot Trans_x a_5 \cdot Rot_x - \pi/2 = \begin{bmatrix} \cos\theta_5 & 0 & -\sin\theta_5 & a_5\cos\theta_5 \\ \sin\theta_5 & 0 & \cos\theta_5 & a_5\sin\theta_5 \\ 0 & -1 & 0 & d_5 \\ 0 & 0 & 0 & 1 \end{bmatrix} \tag{A7}$$

$$M_6^5 = Rot_z\theta_6 \cdot Trans_z d_6 \cdot Trans_x a_6 = \begin{bmatrix} \cos\theta_6 & 0 & -\sin\theta_6 & a_6\cos\theta_6 \\ \sin\theta_6 & 0 & \cos\theta_6 & a_6\sin\theta_6 \\ 0 & 0 & 1 & d_6 \\ 0 & 0 & 0 & 1 \end{bmatrix} \tag{A8}$$

Multiplying these matrices, we obtain a transformation matrix that defines the position and orientation of the $U_6$ gripper system in relation to the $U_0$ basis system.

$$M_6^0 = M_1^0 \cdot M_2^1 \cdot M_3^2 \cdot M_4^3 \cdot M_5^4 \cdot M_6^5 = \begin{bmatrix} n_x & s_x & a_x & p_x \\ n_y & s_y & a_y & p_y \\ n_z & s_z & a_z & p_z \\ 0 & 0 & 0 & 1 \end{bmatrix} \tag{A9}$$

Equation (A9) describes the kinematic model of the presented manipulator.

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
