# Peer review of "Autonomous Manipulator of a Mobile Robot Based on a Vision System"

_applsci, doi:10.3390/app13010439_

Round 1

Reviewer 1 Report

This paper presents a vision system for a manipulator of a mobile robot to operate a panel with switches, perhaps this is an interesting research topic, but the research content of the paper is insufficient, I think this paper is not ready for publication for the reasons but not limited.

There is no innovation in this work due to the proposed vision system not detailed analysis, and the experimental data also the significance of the experimental result are not clear.

The content expressed by the author is too little to prove the scientificity and rationality of the method and experiment.

Author Response

Dear Reviewer 1

The article has been corrected as suggested

This paper presents a vision system for a manipulator of a mobile robot to operate a panel with switches, perhaps this is an interesting research topic, but the research content of the paper is insufficient, I think this paper is not ready for publication for the reasons but not limited.

The article was developed, and a literature review was performed. Based on the reviewed works, it can be seen that there are few applications of marker-based vision systems for manipulators, most of which are based on grasping objects. The analysis of results and conclusions have been improved.

There is no innovation in this work due to the proposed vision system not detailed analysis, and the experimental data also the significance of the experimental result are not clear.

A detailed description of the algorithm is added in Algorithm 1. on page 6. The significance of the experimental results obtained is explained. The novelty of the work is a new application of the vision system for the manipulator to perform differentiated manipulation activities.

The content expressed by the author is too little to prove the scientificity and rationality of the method and experiment.

The research has been clarified and uncertainties clarified. Tests of the accuracy of the designed, programmed and fabricated manipulator, tests of position estimation by the vision system, and a study of the joint operation of the manipulator and vision system prove that with sufficient accuracy the system can perform the tasks assumed by the authors.

Best regards

Authors

Reviewer 2 Report

Brief summary

This study proposes a system for the autonomous operation of a manipulator of a mobile robot. Its main contribution is not clear in the document.

Broad comments

The document is in general hard to follow.

The English needs major spell checking and review.

The document is not well supported with references and should therefore be greatly improved.

The subject of the paper has potential of application, but authors were unable to contextualize and show the importance of the work.

Specific comments

In the Abstract authors should clarify their contribution. It is not clear to the reader if authors developed all the system including the robot or only a part. Please correct and clarify the various parts constituting the developed system.

At the beginning of the Introduction authors should contextualize their work. Please revise.

Authors should include in the Introduction section the related work including references. In line 39 authors briefly reference “recent studies”….

In line 59 authors included the bibliographic reference [10] but there is no reference to 1 to 9 before in the document. Please revise.

In subsection “2.2.1. Marker detection algorithm” authors should include images illustrating the marker detection algorithm step by step.

Authors claim that good lighting is a prerequisite for the proper functioning of the system but never justify this claim or present any results that support it. Authors should include tests with various lighting conditions to support the need for good lighting.

Author Response

Dear Reviewer 2

The article has been corrected as suggested

This study proposes a system for the autonomous operation of a manipulator of a mobile robot. Its main contribution is not clear in the document.

The contribution is a new application of a marker-based vision system for an autonomous manipulator that can perform various manipulation activities. In the literature, the most common application is navigation of mobile robots; solutions for manipulators are less common and focus on grasping objects. The article is supplemented with this information in Chapter 1.

Broad comments.

Unclear issues, such as authors' contributions, example application, literature review, analysis of results, are described more extensively.

The document is in general hard to follow.

Improved the layout of the article, added more explanations and comments on the literature review, the work done, possible applications of the system presented in the article, analysis of the results. Illegible figures were corrected and the operation of the algorithm was presented in paragraphs.

The English needs major spell checking and review.

The language of the article has been corrected.

The document is not well supported with references and should therefore be greatly improved.

A literature review has been done in Chapter 1. More items in the literature have been added.

The subject of the paper has potential of application, but authors were unable to contextualize and show the importance of the work.

A description of possible applications of mobile robots with autonomous manipulators was added in Chapter 1. Such robots will be able to replace humans in various situations, such as industrial work in flexible systems.

Specific comments

In the Abstract authors should clarify their contribution. It is not clear to the reader if authors developed all the system including the robot or only a part. Please correct and clarify the various parts constituting the developed system.

A description of the work performed was added in Chapter 1. The mobile robot, manipulator and vision system were designed, programmed and manufactured by the authors as part of their work in the research club at our university.

At the beginning of the Introduction authors should contextualize their work. Please revise.

The first chapter fills in the missing context of this work.

Authors should include in the Introduction section the related work including references. In line 39 authors briefly reference “recent studies”….

A literature review has been done in Chapter 1. More items in the literature have been added.

In line 59 authors included the bibliographic reference [10] but there is no reference to 1 to 9 before in the document. Please revise.

The order of references has been corrected.

In subsection “2.2.1. Marker detection algorithm” authors should include images illustrating the marker detection algorithm step by step.

The step-by-step algorithm Algorithm 1 has been added.

Authors claim that good lighting is a prerequisite for the proper functioning of the system but never justify this claim or present any results that support it. Authors should include tests with various lighting conditions to support the need for good lighting.

In fact, the article does not provide evidence or tests to prove that a condition for proper operation of the system is good lighting.

Adequate lighting conditions are a basic prerequisite for proper image acquisition. The vision system was tested both under artificial lighting conditions in the laboratory and under outdoor conditions on a sunny and cloudy day. The authors noted the highest number of correct readings under diffuse lighting conditions, shading did little to reduce the effectiveness of the algorithm.

Best regards

Authors

Reviewer 3 Report

In their contribution, the authors focus on the Autonomous manipulator of a mobile robot based on a vision system. The contribution is divided into several consecutive chapters. I have a few questions about the post:

1. The introduction chapter deserves to be expanded and supplemented with references to the literature.

2. On line 66 you have the x, y and z axes. Please write them in italics.

3. Figure 2 deserves better quality.

4. In Figure 3, you probably have text written in purple. This text is illegible.

5. Chapter 4 is focused on discussion. It would be appropriate in this chapter to compare your results with the works of other authors dealing with the issue, if available.

6. Chapter 5 Conclusion deserves to be expanded and better described.

7. Please number the references to literary sources consecutively throughout the post, as they follow each other.

Author Response

Dear Reviewer 3

The article has been corrected as suggested

Comments and Suggestions for Authors

In their contribution, the authors focus on the Autonomous manipulator of a mobile robot based on a vision system. The contribution is divided into several consecutive chapters. I have a few questions about the post:

  1. The introduction chapter deserves to be expanded and supplemented with references to the literature.

A literature review has been done. More items in the literature have been added.

  1. On line 66 you have the x, y and z axes. Please write them in italics.

The font has been corrected.

  1. Figure 2 deserves better quality.

The picture has been corrected.

  1. In Figure 3, you probably have text written in purple. This text is illegible.

Figure 3 has been corrected.

  1. Chapter 4 is focused on discussion. It would be appropriate in this chapter to compare your results with the works of other authors dealing with the issue, if available.

Comparison with other works has been added, however, in other works the studies were similar, but differed in the scale adopted, for example, measurements were conducted for markers with different dimensions measured from different distances. The dimension of the marker and the distance will affect the results obtained. The comparison was made on the basis of relative error, if it was given in another work.

  1. Chapter 5 Conclusion deserves to be expanded and better described.

The conclusions were developed.

  1. Please number the references to literary sources consecutively throughout the post, as they follow each other.

The order of references has been corrected.

Best regards

Authors

Reviewer 4 Report

1、The method in the paper is a general method, and there is no any innovation;

2、The coordinate systems involved need to be defined and diagrammed;

3、The process of detecting of the marker to positioning the manipulator is not described, and it is directly obtained in one sentence. It‘s too colloquial and general.

4、How to measure the actual manipulator movement in enough accuracy of submillimeter class? and how can get the results in accuracy of 0.0001mm and 0.000001deg?

Author Response

Dear Reviewer 4

The article has been corrected as suggested

Comments and Suggestions for Authors

1、The method in the paper is a general method, and there is no any innovation;

The novelty is an innovative application. The method of detecting markers has been applied to a new task - to an autonomous manipulator that can perform various activities. The system has the potential for further development by dispensing with markers and positioning based on objects in the environment, thus increasing its potential for use in unfamiliar environments as well. After reviewing the literature, it can be seen that tags are most often used in the navigation of mobile robots (UGVs) or flying robots (UAVs), while in manipulation it is used less frequently, usually for grasping objects. Corrections have been made to the first chapter.

2、The coordinate systems involved need to be defined and diagrammed;

The coordinate system was adopted in the center of the marker on the left, this was completed in the description and in Figure 3.

3、The process of detecting of the marker to positioning the manipulator is not described, and it is directly obtained in one sentence. It‘s too colloquial and general.

A description of the algorithm in points in Algorithm 1 has been added.

4、How to measure the actual manipulator movement in enough accuracy of submillimeter class? and how can get the results in accuracy of 0.0001mm and 0.000001deg?

Measurements of the performance of the vision system itself were taken on an ABB industrial robot that provides such accuracy. Measurements from the robot's position were taken as a ground truth, and the error of the position calculated by the vision system was counted against it. However, this was corrected and the results were reported to hundredths of a millimeter, as accuracy at this level is sufficient relative to the positioning errors achieved.

Best regards

Authors

Round 2

Reviewer 1 Report

The experimental results are not limited to tabular data, thus, it is suggested to add some visual sample experiments,that would to further improve the readability of the paper.

Author Response

Dear Reviewer
We added some visual sample experiment figures into the paper as suggested (Figure 4.)

“The experimental results are not limited to tabular data, thus, it is suggested to add some visual sample experiments,that would to further improve the readability of the paper.”

Best regards Authors

Reviewer 2 Report

The main issues pointed out in the review were addressed by the authors.

I advise that the paper be accepted for publication.

Author Response

Dear Reviewer

Thank you for your review.

Best regards
Authors

Reviewer 3 Report

Dear authors, thank you for incorporating my suggested comments into your post.

Author Response

Dear Reviewer

Thank you for your review

Best regards
Authors

Reviewer 4 Report

1、There is no innovation in the tag detection algorithm. It is recommended to explain the significance of this article from the whole system.

2、The first test is not the positioning accuracy test, but the execution accuracy, which does not correspond to the statement of line 199. ( The first examined the positioning accuracy of the manipulator)

3、What is the camera's field of view and resolution?

4、What is the actual measuring equipment and method for accuracy testing?

1) How to obtain “the actual manipulator movement” in the first test?

2) How to obtain “the reference distance” in the second test, and how can ABB robot obtain the measurement with an accuracy of 0.01mm? Angle readings and calculated values cannot be taken as measurement accuracy.

3) How to get “the actual distance from the panel mm ” in the third test?

5、The author spent a lot of time explaining the accuracy test results, but did not explain the test equipment and conditions. In addition, what is the accuracy of this test compared with other manipulator with vision systems? The title of this paper is "Automobile manipulator of a mobile robot based on a vision system". It is suggested that the author focus on writing the paper from the perspective of system design, function and application value.

6、There are multiple errors in the serial number of sections in the text.

Author Response

Dear Reviewer Thank you for your review.

Comments and Suggestions for Authors

1、There is no innovation in the tag detection algorithm. It is recommended to explain the significance of this article from the whole system.

Presented vision system is one of many manipulator systems and is used to position the manipulator relative to the panel. It has significant impact on current work of the entire robot system by adding necessary position and orientation corrections into robot control unit. This process is necessary, because of the possibility of contact between the robot and the pressing button. In the case of determining the manipulator distance from the marker, we use the PnP algorithm, and the fact that the marker frame has known dimensions and shape.  In the case of determining the manipulator distance from the marker, we use the PnP algorithm, and the fact that the marker frame has known dimensions and shape. 2D image points and their 3D correspondences are taken only from the outer edges of the marker frame. We also check the thickness of the marker frame and its code. The positions of the markers located on the panel are known relative to global coordinate system, so in PnP algorithm, there could be used all detected markers points.  

2、The first test is not the positioning accuracy test, but the execution accuracy, which does not correspond to the statement of line 199. ( The first examined the positioning accuracy of the manipulator)

We changed mentioned sentence.

3、What is the camera's field of view and resolution?

We added this data in line 126 of the paper. There is also information about camera we used in our system.

4、What is the actual measuring equipment and method for accuracy testing?

1) How to obtain “the actual manipulator movement” in the first test?

We set to our system linear movement for example 50 mm and than we checked the actual movement pose.

Actual manipulator movement was obtained in simple manner. We set some linear movement to manipulator control system and then the end effector actual position was measured. This value was taken from a precision plunger dial indicator RS Pro company. We also put this in the article in 3.1 section.

2) How to obtain “the reference distance” in the second test, and how can ABB robot obtain the measurement with an accuracy of 0.01mm? Angle readings and calculated values cannot be taken as measurement accuracy.

This is 1600 ABB industrial robot and in its specification is written that the accuracy of reaching pose is 0.04 mm. So we changed this in the article in 3.2 section.   

3) How to get “the actual distance from the panel mm ” in the third test?

We use laser sensor di-soric LAT 51 M 500 IG3-B5  (now it is in the paper).

5、The author spent a lot of time explaining the accuracy test results, but did not explain the test equipment and conditions.

We have added a description of the test conditions and the measurement equipment used.

In addition, what is the accuracy of this test compared with other manipulator with vision systems?

We have added more papers to compare results. In the analyzed works, the relative error of position estimation in relation to markers with the use of vision systems usually is about 1% and does not exceed 2%. For comparable tasks, the absolute error was about 0.2 - 3.5 mm. The results always depend on the distance of the camera to the marker and the size of the marker.

The title of this paper is "Automobile manipulator of a mobile robot based on a vision system". It is suggested that the author focus on writing the paper from the perspective of system design, function and application value.

We tried to explain this in introduction section starting from 63 line.

6、There are multiple errors in the serial number of sections in the text.

We changed those errors.

Best regards Authors